# Atypical functional connectivity hierarchy in Rolandic epilepsy

Qirui Zhang [1,2,9], Jiao Li[3,4,9], Yan He[5], Fang Yang[6], Qiang Xu [2,7], Sara Larivière [8], Boris C. Bernhardt [8], Wei Liao [3,4], Guangming Lu [1,2✉] & Zhiqiang Zhang [1,2✉]

Functional connectivity hierarchy is an important principle in the process of brain functional organization and an important feature reflecting brain development. However, atypical brain network hierarchy organization in Rolandic epilepsy have not been systematically investigated. We examined connectivity alteration with age and its relation to epileptic incidence, cognition, or underlying genetic factors in 162 cases of Rolandic epilepsy and 117 typically developing children, by measuring fMRI multi-axis functional connectivity gradients. Rolandic epilepsy is characterized by contracting and slowing expansion of the functional connectivity gradients, highlighting the atypical age-related change of the connectivity hierarchy in segregation properties. The gradient alterations are relevant to seizure incidence, cognition, and connectivity deficit, and development-associated genetic basis. Collectively, our approach provides converging evidence for atypical connectivity hierarchy as a system-level substrate of Rolandic epilepsy, suggesting this is a disorder of information processing across multiple functional domains, and established a framework for large-scale brain hierarchical research.

[1] Department of Diagnostic Radiology, Jinling Hospital, the First School of Clinical Medicine, Southern Medical University, Nanjing 210002, China. [2] Department of Diagnostic Radiology, Jinling Hospital, Nanjing University School of Medicine, Nanjing 210002, China. [3] The Clinical Hospital of Chengdu Brain Science Institute, School of Life Science and Technology, University of Electronic Science and Technology of China, Chengdu 610054, China. [4] MOE Key Lab for Neuroinformation, High-Field Magnetic Resonance Brain Imaging Key Laboratory of Sichuan Province, University of Electronic Science and Technology of China, Chengdu 610054, China. [5] Department of Neurology, Children's Hospital of Nanjing Medical University, Nanjing 210002, China. [6] Department of Neurology, Jinling Hospital, Nanjing University School of Medicine, Nanjing 210002, China. [7] College of Automation Engineering, Nanjing University of Aeronautics and Astronautics, Nanjing 210002, China. [8] Multimodal Imaging and Connectome Analysis Laboratory, McConnell Brain Imaging Centre, Montreal Neurological Institute and Hospital, McGill University, Montreal, QC H3A 2B4, Canada. [9] These authors contributed equally: Qirui Zhang, Jiao Li. ✉email: cjr.luguangming@vip.163.com; zhangzq2001@126.com

Rolandic epilepsy (RE) is the most common pediatric epilepsy and is also known as benign childhood epilepsy with central-temporal spikes[1,2]. This epilepsy manifests with age-dependent phenotype, with an age of onset usually between 4 and 10 years and remittance after adolescence[1,2]. Moreover, it is a gene-associated disease in etiology, sharing common genes, and having high comorbidities with attention-deficit/hyperactivity disorder and autism[3,4]. Thus, RE has been conceptualized as a neurodevelopmental disorder[5]. In clinic, aside from nocturnal convulsive seizures, cognitive deficits in attention, control, and language is the major complaint of children with RE[6]. Such cognitive deficits have been suggested to result either from direct impairments of epileptic incidences[7], or alternatively, was also proposed to be a consequence of delayed brain development[5,8]. Therefore, an overarching system-level developmental imbalance may relate to the co-occurrence of mild seizures and deficits in higher-order cognitive processing. It is essential to clarify the relationship between cognitive abnormalities, epileptic seizures, brain development, and underlying genetic risk factors in Rolandic epilepsy.

Magnetic resonance imaging (MRI) has been the most important neuroimaging tool to investigate neurodevelopment and cognitive functions in health and disease. While most MRI studies have focused on Rolandic areas serving as epileptogenic regions[9–11], a subset of those have related regional and connectivity imaging alterations in the disease to specific cognitive domains[10,11]. For instance, graph theoretical studies have shown alterations in network topology in RE, such as decreased global and regional efficiency[12,13] as well as reduced connectivity and nodal centrality among sensorimotor areas[13], suggested potential functional hierarchical development abnormalities.

In this study, we aimed to answer the following questions: (1) whether children with RE show a typical age-related brain function represented with network hierarchy organization, and (2) whether the age-related abnormalities in brain functions are related to epileptic incidence, cognition, and genetic risk factors. Combined, these behavioral, imaging, and genetic analyses have provided deeper insights into the multiscale mechanism underlying brain development of Rolandic epilepsy.

Previous network findings, however, were analyzed based on separate analysis frameworks, i.e., they all involved networks that were treated at the same level and could hardly reflect the hierarchical characteristics of brain organization. Recently, a novel network analysis strategy, termed connectivity gradients, has been proposed to capture the full extent of a region's participation in the functional organization of the cortex[14,15]. This technique provides comprehensive hierarchical information underlying the segregation and integration of functional brain networks by quantifying gradual transitions between individual regions in the connectivity[15–20]. The principal gradient, which differentiates between transmodal (higher-order/association) and unimodal (sensory) areas, has been widely used to characterize hierarchical cognition processes of working memory, langue and attention[21,22], as well as the abnormalities in brain diseases such as autism[17] and epilepsy[21,23]. In particular, recent studies have shown superiority of connectivity gradients for describing age-dependent shifts in network integration and segregation[22,24]. One of the rather important findings is that principal gradient expand with age and are associated with cognitive development, which is thought to primarily characterize the segregation properties of the connectivity hierarchy[19,22,24–26].

In addition to histogram variation of each gradient, here we calculated a metric of gradient eccentricity to quantify the overall brain network separation trend with age based on three main gradients. Gradient eccentricity values were computed as the Euclidean distance between each vertex and the centroid of all vertices in each aligned individual 3D gradient space[25]. Based on this metric we can quantify RE age-related alterations in brain network segregation properties of the connectivity hierarchy. We further used spatial transcriptomic association analysis and various types of enrichment analysis to advance our understanding of the relationship between age-related eccentricity changes and developmental process or molecular mechanisms, based on the Allen Human Brain Atlas[27].

In the present study, by measuring multi-dimensional functional connectivity gradients and characterizing the gradients with an eccentricity metric, we investigated alteration features of age-related connectivity hierarchy in RE. (1) RE presented contracted functional connectivity, i.e., decreased gradient eccentricity, in the unimodal of rolandic regions and occipito-temporal area, and connectivity expansion in low-order transmodal of superior parietal lobule; the gradient alterations were relevant to seizure incidence, cognition, and connectivity deficit in the patients. (2) RE showed decreased age-related change of connectivity, indicating an atypical age-related connectivity hierarchy in segregation properties. The more rapidly hierarchical organization changed in typically developing children (TDC), the greater decreased age-related change in RE patients occurred. Combining with spatial transcriptomic association analysis, we further found (3) The genes basis of the gradient alterations was associated with spatial-temporal brain development characteristics, developmental biological processes, synapses, and developmental-related diseases. Collectively, our approach provides converging evidence for atypical connectivity hierarchy as a system-level substrate of Rolandic epilepsy, suggesting Rolandic epilepsy is a disorder of information processing across multiple functional domains.

## Results

**Demographic and clinical characteristics.** A total of 140 children with RE and 91 TDC were included in the analysis after excluding 22 pateints and 26 TDCs through data quality control (Supplementary Fig. 1). Table 1 summarized the demographic and clinical characteristics of all subjects. No significant differences in age, sex, and relative fMRI head motion were observed between patients and controls.

**Age-related changes in hierarchy organization.** Surface-based, vertex-wise functional connectivity gradients were generated from preprocessed fMRI data using BrainSpace[15–17]. The first three gradients, as chosen for eccentricity analysis, accounted for

**Table 1 Demographics and clinical data.**

|  | RE | TDC | t/$\chi$2 | p statistic |
|---|---|---|---|---|
| No. of subjects | 140 | 91 | _ | _ |
| Sex (Male / Female) | 70/70 | 50/41 | 0.54 | 0.46 |
| Age (years) | 9.22 ± 2.28 | 9.28 ± 2.26 | 0.29 | 0.76 |
| Age at seizure onset (years) | 7.92 ± 2.41 |  |  |  |
| Epilepsy duration (months) | 18.50 ± 14.02 |  |  |  |
| Seizure times | 2.60 ± 0.89 |  |  |  |
| Seizure free duration (months) | 4.48 ± 8.99 |  |  |  |
| AED (naive/medication) | 77/63 |  |  |  |
| Relative fMRI head motion (mm) | 0.11 ± 0.05 | 0.12 ± 0.05 | 0.60 | 0.54 |

*TDC* typically developing children, *RE* Rolandic epilepsy, *AED* anti-epileptic drugs.

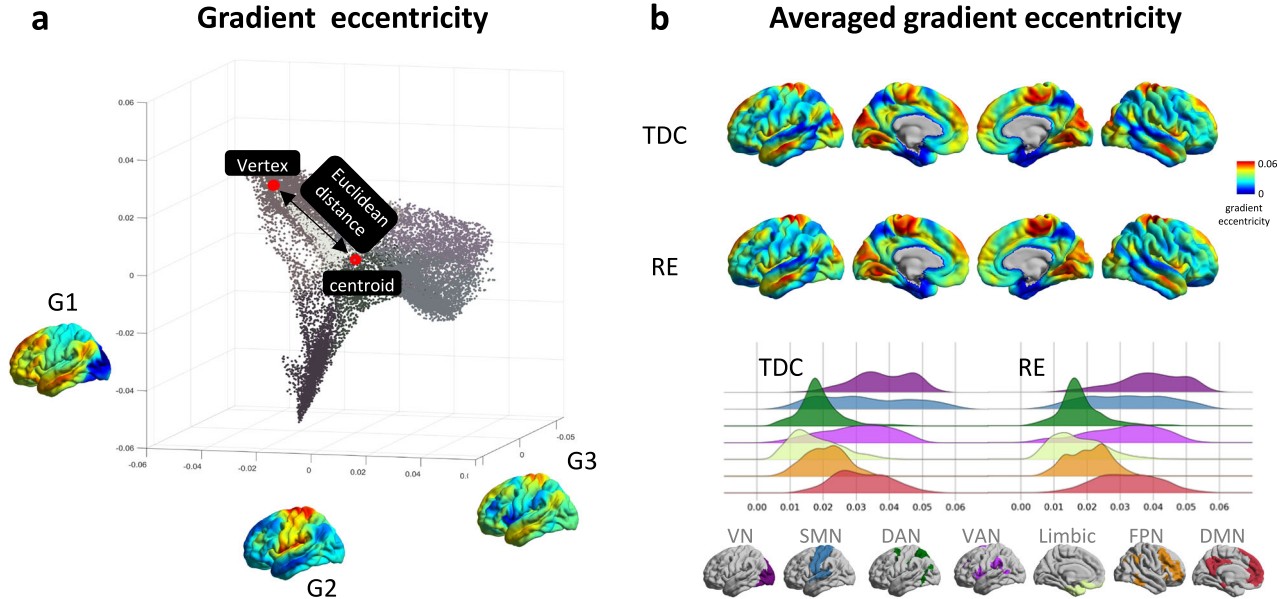

**Fig. 1 Functional connectivity gradient eccentricity. a** Gradient eccentricity was calculated as the Euclidean distance between the centroid of all vertices and each vertex in an individual 3D gradient space. **b** Brain surfaces map showed mean gradient eccentricity values of RE (n = 140) and TDC (n = 91), joy plots showed relatively high eccentricity in VN, SMN, and DMN. TDC typically developing children, RE Rolandic epilepsy, VN visual network, SMN sensorimotor network, DAN dorsal attention network, VAN ventral attention network, FPN frontoparietal network, DMN default mode network.

44.22% of the variance. In TDC, averaged gradients showed functional differentiation running from sensorimotor network (SMN)-to-default mode network (DMN; G1), DMN-to-visual network (VN; G2), and SMN-to-ventral attention network (VAN; G3; Supplementary Note 1 and Supplementary Fig. 2), thus reflecting functional discrimination across different levels of cortical hierarchy. For each gradient, we performed spatial correlations between the mean gradients and age-related changes (statistically assessed by spin permutation) and showed that the gradients tend to be dispersed along with age increase (Supplementary Note 1 and Supplementary Fig. 2). Based on this pattern, we used the gradient eccentricity to assess dispersion properties of the multi-dimensional hierarchical organization. Gradient eccentricity values were computed as the Euclidean distance between each vertex and the centroid of all vertices in each aligned individual 3D gradient space. Higher values in gradient eccentricity were distributed in the VN, SMN, and DMN. These networks were located at the apex of each axis in the 3D gradient space (Fig. 1). For correlation between gradient eccentricity and age, only positive coefficients were discovered in either vertex-wise, community-wise, and cortical average-wise analysis, mainly affecting regions in the VN, VAN, frontoparietal network (FPN), and DMN (Fig. 2a, Supplementary Note 2 and Supplementary Fig. 3). Moreover, robust results were achieved after controlling for possible imaging covariates of functional connectivity density (FCD) or cortical thickness (Supplementary Note 2, Supplementary Fig. 3). No significant sex × age interaction effect was observed on eccentricity (Supplementary Note 3, Supplementary Fig. 4)[28]. Both results implicate that the gradient eccentricity changes are specifically related to age.

**Age-related alterations of connectivity gradients in Rolandic epilepsy.** Using vertex-wise disease × age interaction analysis, we first observed age-related gradients alterations in RE relative to TDC. Significant alterations were observed in occipito-temporal areas (G1), hand motor area, occipito-temporal areas, and superior temporal gyrus (G2), superior parietal lobule, mouth motor area, and supplementary motor area (G3; Fig. 2d).

Functional community analysis further showed network-specific alterations in the VN (G1), SMN (G2), and dorsal attention network (DAN, G3; $q_{FDR} < 0.05$; Supplementary Note 4, Supplementary Figs. 5 and 6).

In the analysis of the interaction between disease and age, the regression coefficients of the dominant eccentricity in the superior parietal lobule, primary motor area and occipitotemporal area were decreased in children with RE. ($p_{FWE} < 0.05$; Fig. 2c). Functional community analysis highlighted this significant decrease in VN ($q_{FDR} < 0.05$), and marginal decrease in the DAN ($p_{uncorrected} < 0.05$; Fig. 2e). Reproducibility analysis revealed that the spatial patterns of disease × age interaction were stable across different sample sizes, as assessed by bootstrapping analysis (Supplementary Note 5, Supplementary Fig. 7). No AED (Antiepileptic drug) × age interaction effect was observed when estimating the possible AED effect on eccentricity age-related trends (Supplementary Note 6, Supplementary Fig. 8).

Cross community correlation analysis showed a significant association between typical and atypical age-related eccentricity changes. Disease × age interaction t-map was significantly correlated with age-related changes across different networks (Pearson's $r = -0.76$, $p = 0.04$), indicating that rapidly developing networks tend to show severer age-related alteration (Fig. 2f).

**Disease effects on connectivity gradients.** Case-control t-test revealed increases of eccentricity in the superior parietal lobule, primary motor cortex, and decreases in the occipito-temporal and centro-temporal areas (Rolandic region) in Rolandic epilepsy (Fig. 3a). Functional community analysis highlighted the increases of eccentricity in the DAN and decreases in the VN ($q_{FDR} < 0.05$; Fig. 3b). Consistent results were also found in case-control t-test in each gradient (Supplementary Note 4, Supplementary Fig. 6).

Compared to TDC, patients present decreased FCD in the precuneus, medial prefrontal lobe, temporal pole, superior parietal lobule, dorsal prefrontal lobe, and supramarginal gyrus. Significant clusters were located in the DMN (25.0% of significant vertex), VAN (23.6%), and limbic network (19.23%).

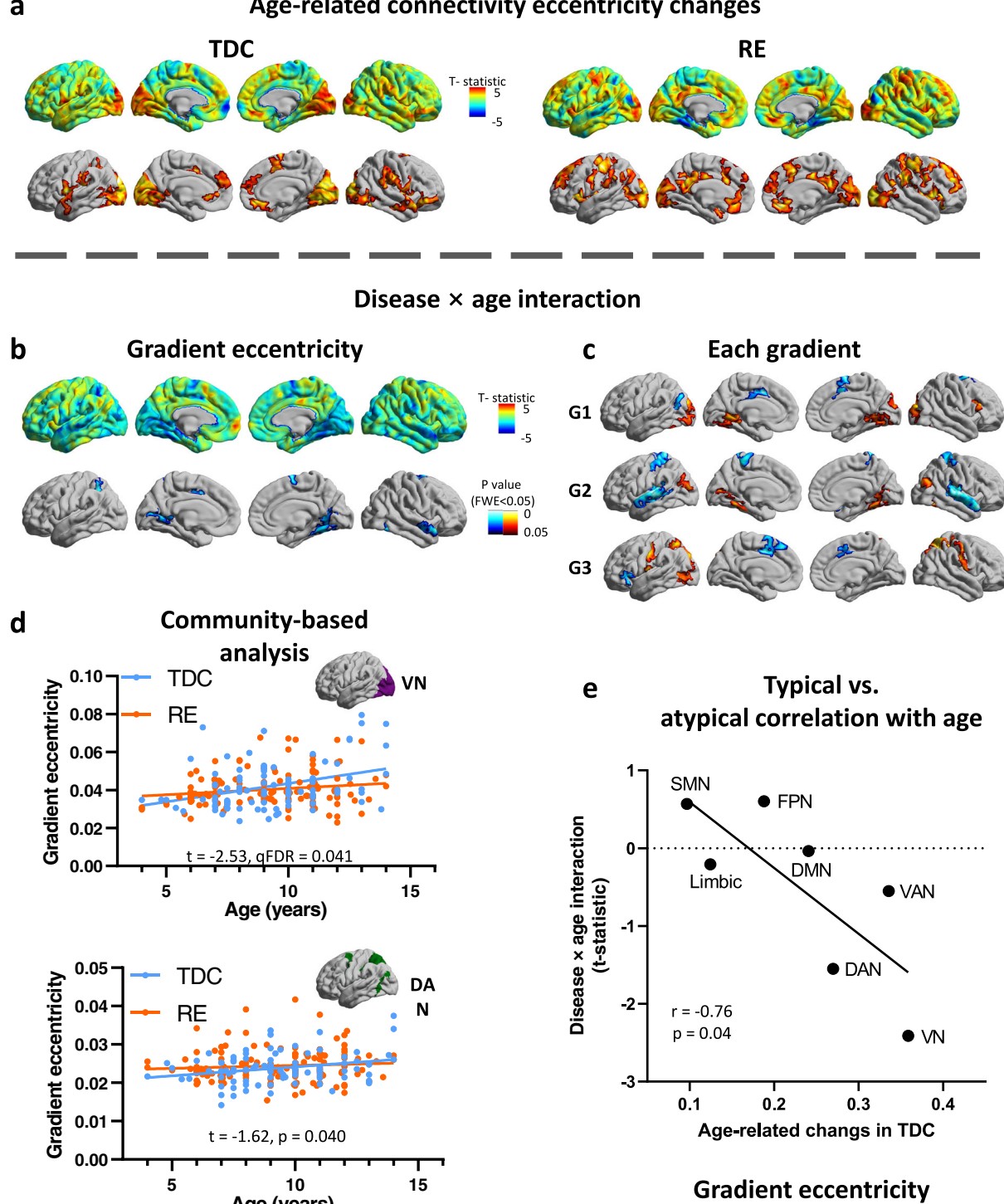

**Fig. 2 Atypical age-related gradient eccentricity in Rolandic epilepsy. a** Surface-based age-related eccentricity changes in Rolandic epilepsy ($n = 140$) and TDC ($n = 91$). **b** Surface-based disease × age interaction analysis ($n = 231$), with significant decrease age-related change in superior parietal lobule, primary motor area and occipito-temporal area. **c** Surface-based disease × age interaction in each gradient ($n = 231$). **d** Community-based disease × age interaction of gradient eccentricity ($n = 231$) shows significant decrease age-re in the VN and DAN. **e** The relationship between typical and atypical age-related gradient eccentricity. A community-based scatterplot ($n = 7$ communities) of the age-related changes in TDC (x-axis) and disease × age interaction t-map (y-axis). TDC typically developing children, RE Rolandic epilepsy, VN visual network, SMN sensorimotor network, DAN dorsal attention network, VAN ventral attention network, FPN frontoparietal network, DMN default mode network.

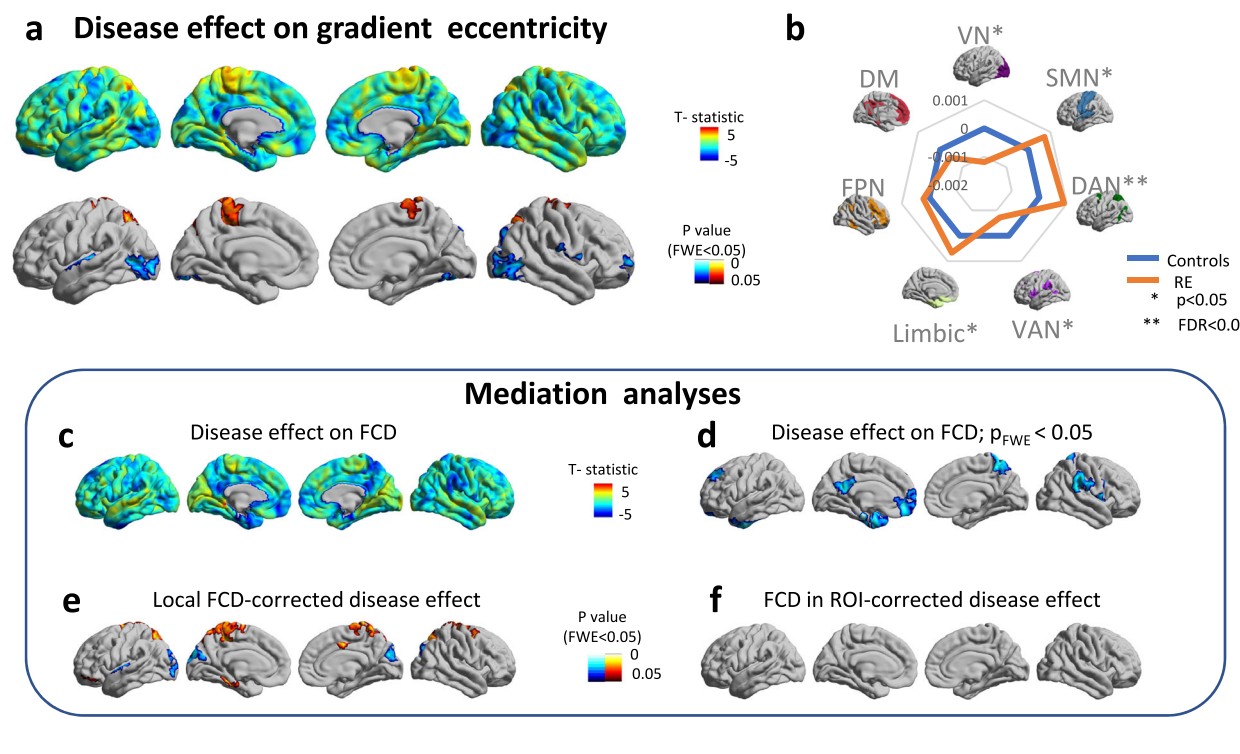

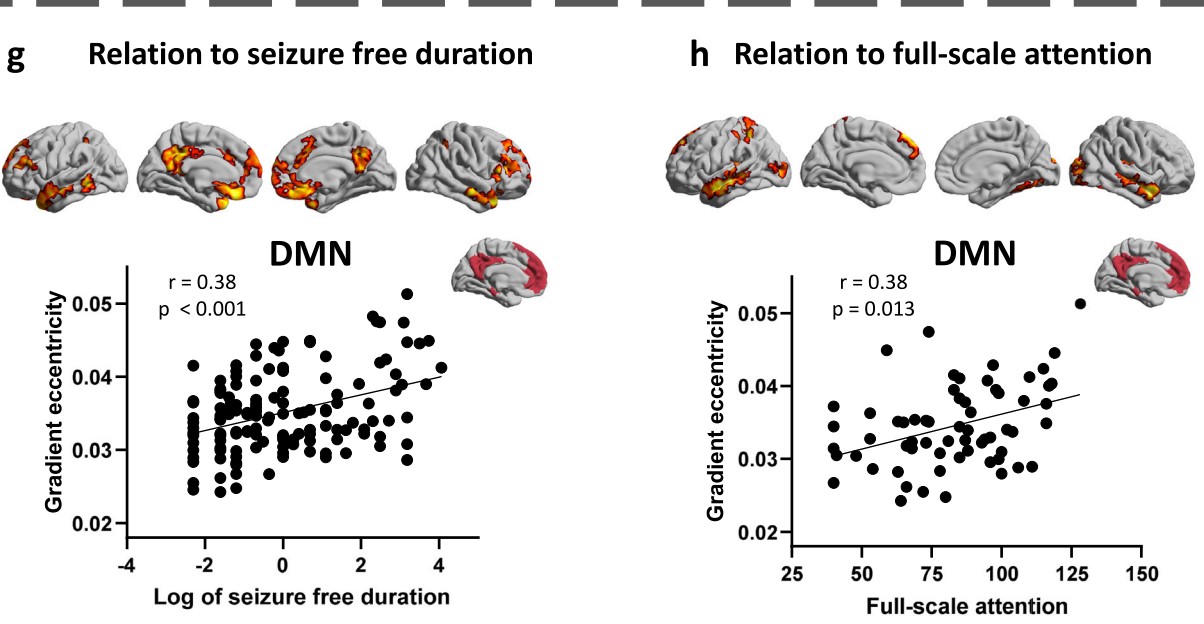

**Fig. 3 Gradient eccentricity mapping in Rolandic epilepsy and TDC and relation to relation to cognitive and clinical variable. a** Surface-based case-control t-test between Rolandic epilepsy and TDC ($n = 231$), with significant increases in the superior parietal lobule, primary motor cortex, together with decreases in occipito-temporal and centro-temporal areas (Rolandic region). **b** Community-based z-score analysis ($n$ = number of vertices in each community) of gradient eccentricity (with respect to controls) shows significant increases primarily in DAN, together with decreases in primarily in VN. **c, d** Compared to TDC, patients present decreased FCD in the precuneus, medial prefrontal lobe, temporal pole, superior parietal lobule, dorsal prefrontal lobe, and supramarginal gyrus. **e** Findings of eccentricity alterations remained robust after correcting for FCD in each vertex. **f** But the effect size of eccentricity alterations in Rolandic epilepsy was markedly reduced when analyses were repeated while controlling for mean FCD of the significant region in DMN (right bottom). **g** Surface-based and community-based (DMN) correlation between eccentricity and log of seizure free duration in RE ($n = 140$). **h** Surface-based and community-based (DMN) correlation between eccentricity and attention in RE ($n = 65$). TDC typically developing children, RE Rolandic epilepsy, VN visual network, SMN sensorimotor network, DAN dorsal attention network, VAN ventral attention network, FPN frontoparietal network, DMN default mode network, FCD functional connectivity density.

In mediation analyses, findings of eccentricity alterations remained robust after correcting for FCD at each vertex, suggesting independence from local connectivity. Effect sizes of eccentricity alterations in Rolandic epilepsy, however, were markedly reduced when analyses were repeated while controlling for mean FCD of the significant region in DMN (effect size reductions up to 25% across clusters of findings, with strongest reductions in occipital lobe), suggesting a mediatory role of

connectivity deficit in DMN on the hierarchical organization (Fig. 3c).

For correlation analyses between eccentricity and clinical variables, positive correlations were observed between gradient eccentricity and seizure free duration mainly in the precuneus, medial prefrontal cortex, and temporal pole (accounting for 52.3% in significant vertex for DMN), but no correlation was found between eccentricity and seizure times (Fig. 3d).

In addition, A subset of children in our database performed simultaneous EEG-fMRI ($n = 48$) as previous described[11,29], and it can be used to monitor for obvious intermittent epileptic activity during the scan (30 patients with discharge and 18 without discharge). We found there is no significant difference between two sub-groups. Detailed methods and results can be found in the supplementary material (Supplementary Note 7).

Correlation analyses were performed between eccentricity and Raven's total score as well as full-scale response control. Positive correlations were observed between gradient eccentricity and full-scale attention and mainly affected the temporal pole, postcentral areas, and occipito-temporal areas (accounting for 38.6% in significant vertex for DMN). We did not find an association between gradient eccentricity and Raven's total score, nor full-scale response control (Fig. 3e).

**Altered gradient eccentricity and cortical gene expression in Rolandic epilepsy**. We performed spatial correlation analysis between age-related alterations of gradient eccentricity in children with Rolandic epilepsy and whole-brain gene expression using transcriptomic data from the AHBA[30] (http://human.brain-map.org). Partial least squares regression (PLS) was used to determine the spatial relationship between hierarchical age-related alterations and transcriptional activity. In the PLS analysis, the first component (PLS1) represents the spatial map that captures the greatest (19.4%) fraction of total gene expression variance across cortical areas ($p_{spin} < 0.0001$). The distribution of the PLS1 weights reflected an anterior-posterior pattern of gene expression. And PLS1 weighted gene expression map was spatially correlated with the disease × age interaction t-map (Pearson's $r$ (146) = 0.44, $p_{spin} < 0.0001$; Fig. 4a), which is interpreted as an areal variation in the transcriptional architecture of the human cortex that is also captured in the hierarchical age-related alteration of Rolandic epilepsy in the gradient eccentricity map[31]. We ranked the normalized weights of PLS1 based on univariate one-sample $Z$ tests and found that 1627 PLS genes ($q_{FDR} < 0.05$) explained the age-related change in the eccentricity gene list in Rolandic epilepsy patients.

Enrichment analyses were performed for the list of most strongly associated genes ($n = 1627$, $q_{FDR} < 0.05$). Using the cell-type-specific expression analysis tool[32], developmental gene set enrichment analysis compared the selected gene list with developmental enrichment profiles. This analysis highlights developmental time windows across macroscopic brain regions in which genes are strongly expressed. We found marked expression of the genes enriched from childhood onward in the cortex, thalamus, and cerebellum during the childhood-to-adolescence time window (Fig. 4b).

Next, we tested whether the transcriptional features associated with gradient eccentricity alterations captured relevant neuro-pathological information. In this TopGene analysis, the top significant GO: biological process were "neuron differentiation", "neuron development", "plasma membrane bounded cell projection organization", "generation of neurons" and "cell projection organization". These terms are relevant to neuron development and projection. Top significant GO: cellular component mainly included synapses and their substructures. In addition, genes related to "epilepsy", "Intellectual Disability", "Schizophrenia", "developmental delay" showed the strongest overlap with the gene list (Fig. 4c).

**Discussion**
We described connectivity hierarchy organization of gradients by quantified the diffusivity of vertices in three-dimensional connectivity gradients space. Recent studies have focused on the development of a single gradient in children and adolescents. The gradients constructed in this study were highly consistent with the spatial pattern of 9 years old in Dong et.al. work[24], and the trend of principal gradient expansion with age of was consistent with recent work[19,22]. This paper confirms that gradients were dispersed with age, and the eccentricity showed a monotonic increase trend with age. These robust findings allowed us to use the gradients eccentricity to synthesize the brain network separation trend with age of multi-dimensional gradients. This approach provided a good tool for the investigation of developmental-related brain diseases. Like the cortical thickness, gradients eccentricity could describe the age-related trajectory in a simply linear way, which made it possible to describe age-related alteration by interaction analysis.

Compared with TDC, Rolandic epilepsy patients showed contracted connectivity in rolandic region, which is the epileptogenic region[9–11]. In addition to structural[9,33] and functional[10,13] changes, our results suggested there are also hierarchical abnormalities in rolandic region, highlighted the vital role of the regions in the processing of disease. Another region showing contracted connectivity, i.e., the occipito-temporal area, is engaged in the integration of feedforward and feedback streams, and particularly plays important roles in facial recognition[34] and language processing[35]. Gradient changes in the occipito-temporal areas were also found in other neurodevelopmental disorders, such as autism[17], and may be correlated with face motor symptoms[1,2] and language deficits in these diseases[6]. In addition, the pattern of connectivity expansion was found in low-order transmodal of superior parietal lobule, as the foremost node in DAN. These findings suggested that Rolandic epilepsy also affected DAN in low order transmodal, consistent with the characterized attention deficits in Rolandic epilepsy patients[36,37]. DMN, as the high-order transmodal, engaging in multilevel neural information processing, was found to be correlated with seizure-free duration and attentional behavior in the patients. Epileptiform discharges and seizure activity associated with cognitive deficits (e.g., attention) in individuals with Rolandic epilepsy[4,38] by inhibiting the DMN[10,39]. These findings indicated that the hierarchy of DMN also played an important role in epilepsy remission and cognition improvement in epileptic children. We also found that the hierarchical organization alterations in Rolandic epilepsy was mediated by DMN connectivity deficits[40], suggesting cascading effects of the DMN on connectivity hierarchy in Rolandic epilepsy.

By unraveling an age-related monotonic increase trend of eccentricity in TDC, we first validated an optimal imaging parameter for depicting brain development by synthesizing comprehensive properties of multi-dimensional gradient[22]. Children with Rolandic epilepsy presented an atypical pattern of age-related trend of eccentricity, particularly in regions involved in disease effect analysis (occipito-temporal area and superior parietal lobule). In typically developing individuals, vision was the network with the fastest age-related eccentricity change, and its age-related alteration can represent the deficits of whole brain hierarchical organization. These results indicated that the cognitive function deficits in Rolandic epilepsy, represented with network hierarchy, was an effect of abnormal brain development.

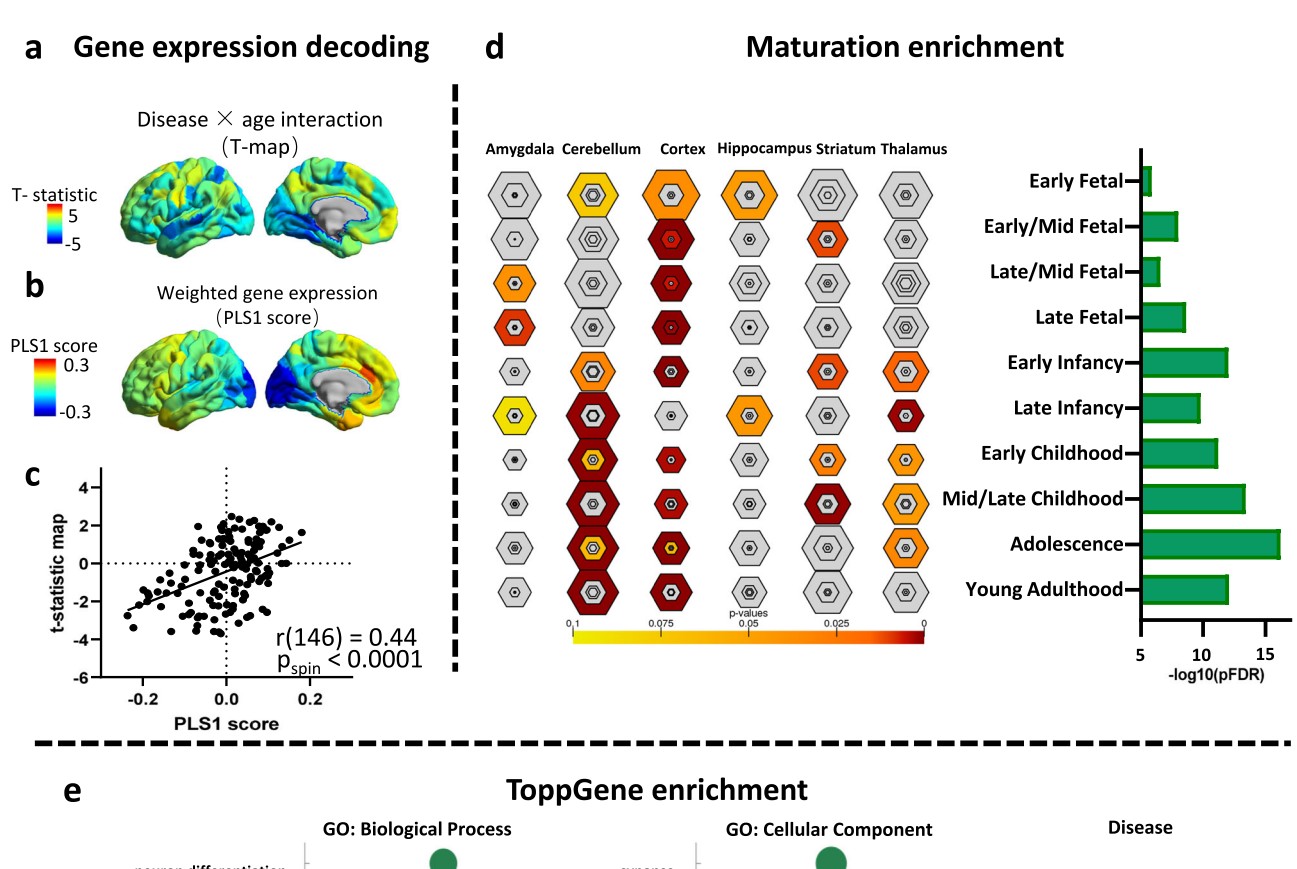

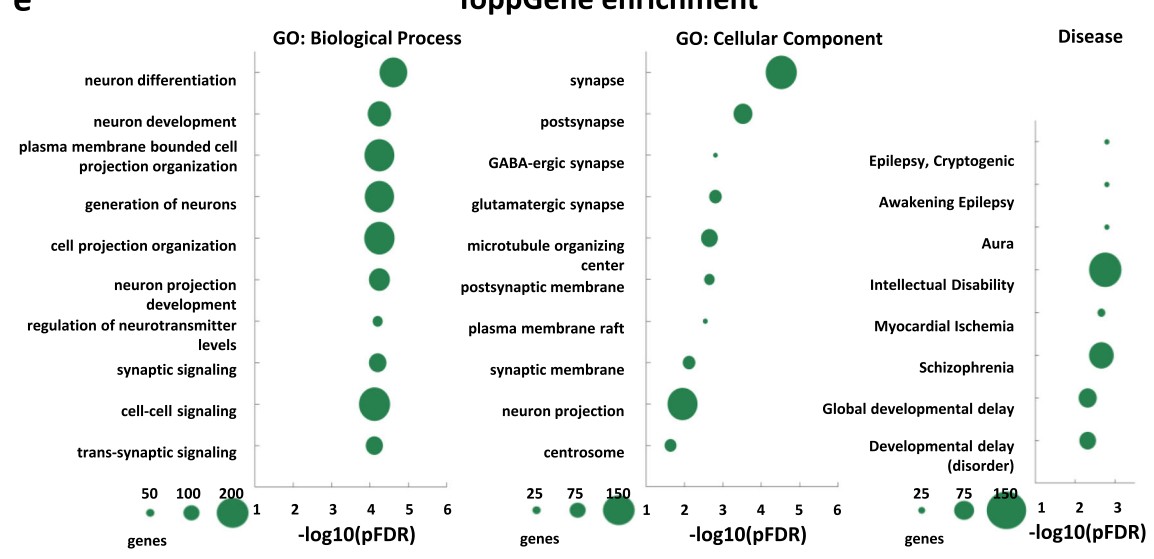

**Fig. 4 Transcriptomic analysis and development enrichment. a** Gene expression decoding of disease × age interaction. Statistic value in the left hemisphere. **b** A weighted gene expression map of regional PLS1 scores in the left hemisphere. **c** A scatterplot of regional PLS1 scores (a weighted sum of 10,027 gene expression scores) and disease × age interaction (Pearson's r (146) = 0.44, $p_{spin}$ < 0.0001) (bottom). **d** We identified a gene set (n = 1627 genes) that showing consistent whole-brain expression pattern. These genes were input to a developmental enrichment analysis, showing strong associations with the cortex, thalamus, and cerebellum during the childhood-to-adolescence time window. **e** Enriched ontology terms in GO (gene ontology): biological process, GO: cellular component, and disease for the gene set. The size of the circle represents the number of genes involved in each term.

This work first showed atypical age-related functional alteration and relevant factor of Rolandic epilepsy, further quantitative analysis may be helpful to quantify the degree of delayed functional brain development. E.g., structural MRI studies have demonstrated a 0.45-year structural development delay by deep learning-based brain age prediction[9]. Interestingly, a phenomenon was found that the networks showing greater changes in Rolandic epilepsy were overlapped with those having rapidly developed in TDC. It suggests that the rapid development of system-level brain hierarchies is disrupted by Rolandic epilepsy.

Above findings were also independently supported by transcriptomic association studies and developmental enrichment analyses. We found age-related changes of gradient eccentricity were associated with genes expressed cortex, thalamus, and cerebellum during late childhood-to-adolescence time window. It is consist with Rolandic epilepsy related brain region and self-limited natural history[2], and contributed evidence in microscale and macroscale levels for understanding the spatial-temporal characteristics underlying neural mechanisms of Rolandic epilepsy. In addition, GO biological processes enrichment in

TopGene analysis showed that gene set relevant to annotations associated with neuron development and projection. Previous studies have also shown cortical structural changes and neuronal projection disorders in Rolandic epilepsy[9,33]. Our results provide further evidence for the biological basis of the changes in brain development of Rolandic epilepsy. The main enriched annotations in GO: cellular component were synapses, especially excitation and inhibition synapses. A hypothesis has been proposed that childhood epilepsy resulted from a dysregulating synaptic activity at a critical developmental period in the immature brain[41,42]. During this critical period, the brain's neural architecture is sharply remodeled, involving synaptic elimination, dendritic pruning, and myelination[43]. Coupled network and molecular changes may ultimately change cortical circuit properties, including the balance of excitation and inhibition (E/I). Previous studies considered imbalances in cortical E/I as an important pathogenesis of neurodevelopmental disorders[44]. Brain network E/I imbalance is also found in Rolandic epilepsy[45]. We assumed that E/I imbalance induced by abnormal synapses related gene expression was involved in the age-related alteration of Rolandic epilepsy. Furthermore, our disease enrichment showed that genes in "epilepsy", "Intellectual Disability", "Schizophrenia", "developmental delay" had the strong overlap with Rolandic epilepsy. Together with evidence that Rolandic epilepsy presents comorbidities of attention-deficit hyperactivity disorder and autism[4,5], these findings supported that Rolandic epilepsy share a common mechanism with common neurodevelopmental disorders[3,4].

Several limitations are noteworthy. First, although with a rather large size of population, the study was based on single center, cross-sectional dataset. The observed differences in functional connectivity hierarchy are indicative of age-related alteration in Rolandic epilepsy, rather than definitive evidence of developmental changes. Future study is needed with follow-up and multi-center design to confirm these findings and investigate the potential changes in connectivity hierarchy over time. Second, we only adopted a normative analysis based on the AHBA, analysis of biological samples from patients themselves may provide more solid evidence. We also note that the combination of virtual brain and clinical and developmental factors through the mechanistic model may offer us the possibility to answer the casualty relationship between development, epilepsy, and genes[46,47].

In conclusion, we not only established a path and framework for large-scale brain hierarchical research but also suggested atypical connectivity hierarchy as a system-level substrate of Rolandic epilepsy. Furthermore, we promoted a comprehensive understanding of the relevant microscale processes.

## Methods

**Participants**. A cohort of 162 children diagnosed with Rolandic epilepsy was recruited from Jinling Hospital between 2015 and 2020. The sample consisted of 81 boys and 81 girls, aged between 4 and 14 years, with a mean age of $9.16 \pm 2.32$ years (mean ± SD). The diagnosis of Rolandic epilepsy was made by two experienced neurologists (Y.H. and F.Y.) following the classification criteria established by the International League Against Epilepsy (ILAE)[48]. In addition, a group of 117 typically developing children (TDC) without a history of neurological or psychiatric disorders, comprising 65 boys and 52 girls, aged between 4 and 14 years, with a mean age of $9.43 \pm 2.56$ years, were recruited from local schools. All participants underwent routine structural MRI, and the results revealed normal radiological findings. Furthermore, cognitive psychological assessments were conducted on 65 patients, including Raven's Standard Progressive Matrices (RSPM) and the Integrated Visual and Auditory Continuous Performance Test (IVA-CPT)[49].

This research was approved by the medical ethics committee in Jinling Hospital, Nanjing University School of Medicine, and written informed consent was obtained from the guardian of each participating children.

**Imaging acquisitions**. All participants underwent functional and anatomical data acquisitions using a 3 T MRI scanner (SIEMENS Trio Tim, Siemens Healthcare) by the same protocol described in previous studies[9,50]. To minimize head movements, individuals were instructed to keep their eyes closed and remain awake during the scan, with a foam padding placed between their head and the coil. Functional MRI (fMRI) data were obtained using a single-shot echo-planar imaging sequence with the following parameters: repetition time of 2000 ms, echo time of 30 ms, field of view of $240 \times 240$ mm$^2$, in-plane matrix of $64 \times 64$, and a flip angle of 90°. Each child underwent a total scan time ranging from 500 to 2000 seconds, during which 30 transverse slices (slice thickness of 4 mm and interslice gap of 0.4 mm) were acquired, aligned along the anterior-posterior commissure line. In addition, high-resolution 3D T1-weighted anatomical images were acquired using a magnetization-prepared rapid gradient-echo sequence with the following parameters: repetition time of 2300 ms, echo time of 2.98 ms, flip angle of 90°, field of view of $256 \times 256$ mm$^2$, and slice thickness of 1 mm.

**Imaging processing**. Imaging processing procedures for all participants was implemented by the same procedures described in previous studies[51–53]. Anatomical (T1-weighted) data underwent processing using FreeSurfer (v6.0.0, http://surfer.nmr.mgh.harvard.edu). The sequence of steps for pre-processing fMRI data is as follows: (1) correction for slice timing, (2) rigid body correction for head motion, (3) normalization of global mean signal intensity across runs, (4) application of a bandpass filter (0.01–0.08 Hz), and (5) regression of nuisance variables (including six motion parameters, white-matter signal, ventricular signal, whole-brain signal, and their temporal derivatives). Following preprocessing, the fMRI data were linearly aligned to each participant's corresponding high-resolution anatomical images through boundary-based registration. Subsequently, the functional images were registered to the FreeSurfer cortical surface template (fsaverage6). To improve signal quality, a 6-mm full-width half-maximum (FWHM) smoothing kernel was applied to the fMRI data on fsaverage6 and subsequently down-sampled to the fsaverage5 template (10,242 vertices per hemisphere). A detailed quality control process is provided in Supplementary Fig. 1. Briefly, structural data with poor quality and fMRI data displaying significant head motion were excluded. Finally, 500 s of fMRI data with the least head motion were selected for subsequent analysis.

**Gradient and eccentricity analyses**. Surface-based, vertex-wise functional connectivity gradients were generated from preprocessed fMRI data using BrainSpace[15–17]. First, in each subject, vertex-based functional time-series were used to create functional connectivity matrices using pairwise Pearson correlation. Connectivity matrices were subjected to row-wise thresholding (top 10% of edges maintained). We then calculated the cosine distance between all pairs of rows to estimate the similarity in connectivity patterns between each pair of voxels, thereby obtaining a symmetrical similarity matrix. We applied diffusion embedding with a manifold learning parameter of $\alpha = 0.5$ to the symmetrical similarity matrix to identify multiple lowdimensional gradients to resolve the gradients of subject-level connectivity. In brief, the algorithm estimates a low-dimensional embedding from a high-dimensional connectivity matrix. In this space, cortical vertices that are strongly interconnected by either many connections or few very strong connections are closer together, whereas vertices with only little or no inter-connectivity are farther apart. A group-averaged functional connectivity matrix of TDC was also constructed, thresholded, normalized, and subjected to diffusion map embedding. The embedding solution from each subject were then aligned to the group-level embedding via procrustes rotations.

To abstract overall brain network separation trend from multi-dimensional features of functional connectivity gradients, we calculated gradient eccentricity, a 3D gradient dispersion property[25]. Gradient eccentricity values were computed as the Euclidean distance between each vertex and the centroid of all vertices in each aligned individual 3D gradient space. We averaged gradient eccentricity values in each group, and compared their distributions in each functional community (Yeo 7 networks[54]).

**Age-related alterations and disease effects of connectivity gradients**. We used vertex-wise linear models implemented in SurfStat (https://math.mcgill.ca/keith/surfstat)[55] to compare patterns of age-related gradient changes and gradient eccentricity changes in each group. Sex and head motion were set as regressors in the models. Spatial correlation was used to assess the relationship between age-related changes and mean TDC value. We also applied a well-established functional community[54] decomposition to summarize our surface-wide findings. To determine the unique contribution of eccentricity versus FCD or cortical thickness to age, we ran robust linear regression for each functional community on eccentricity controlling for FCD or cortical thickness respectively, of the same network (and vice versa), using following linear models:

$$\text{Age} \sim 1 + \text{Eccentricity} + \text{FCD} + \text{Sex} + \text{Head Motion} + e \qquad (1)$$

$$\text{Age} \sim 1 + \text{Eccentricity} + \text{Cortical thickness} + \text{Sex} + \text{Head Motion} + e \qquad (2)$$

Global FCD mapping was calculated to measure the vertex-wise functional connectivity strength[56]. The number of functional connections was determined through Pearson correlations between time-varying signals of a vertex and those in other vertex using an arbitrary threshold $r = 0.3$.

Connectivity gradients in children with Rolandic epilepsy were compared to TDC. age-related alterations were investigated using a disease × age interaction, and disease effect was investigated using case-control *t*-tests. Statistical tests for both gradient and gradient eccentricity analyses were calculated using surface-based linear models implemented in SurfStat. Models included effects of age, sex, head motion, and group. We also applied functional community decomposition to summarize surface-wide findings.

To examine Rolandic epilepsy-related hierarchical abnormalities beyond the effects of FCD, surface-based linear models first compared FCD in Rolandic epilepsy relative to TDC while controlling for age, sex, and head motion. Mean FCD values of significant clusters was extracted. We then assessed gradient eccentricity alterations in Rolandic epilepsy while controlling for FCD value at each vertex or FCD in significant cluster.

Associations between hierarchical brain organization and clinical feature or cognition features in Rolandic epilepsy patients were explored. Clinical features including seizure-free duration or seizure times. The seizure-free duration was normalized by taking logarithms. Epilepsy duration and age at seizure onset were not included in this analysis as they were highly correlated with age (epilepsy duration: $r = 0.24$, $p = 0.034$; age at seizure onset: $r = 0.79$, $p < 0.001$). Cognition features including Raven's total score, full-scale response control, and full-scale attention in patients who successfully underwent cognitive psychological assessments. Correlation analyses were carried out between gradient eccentricity and features using SurfStat, age and sex were included in the regression models as covariates.

**Age-related alterations and gene expression**. We performed spatial correlation analysis between age-related alterations of gradient eccentricity in children with Rolandic epilepsy and whole-brain gene expression using transcriptomic data from the AHBA[30] (http://human.brain-map.org). The processed cortical gene expression provided by Arnatkevic et al.[57], 10,027 selected gene were assigned to a parcellation scheme of 300 parcels[58]. Because the AHBA dataset includes right hemisphere data in only two brains (out of six postmortem brains), only hemisphere data (150 regions × 10,027 gene expression levels) was used in our analysis. Partial least squares regression (PLS) was used to determine the relationship between hierarchical age-related alterations (mean t-values of disease × age interaction of gradient eccentricity from 150 cortical regions in the left hemisphere) and transcriptional activity for all 10,027 genes. The first component of the PLS (PLS1) was the linear combination of gene expression values that most strongly correlated with age-related alterations. Spin permutation testing based on spherical rotations of the t-value map (5000 times)[59] was used to test the null hypothesis that PLS1 explained more covariance between the gradient eccentricity map and whole-genome expression than expected by chance[60]. Bootstrapping was used to estimate the variability of each gene's PLS1, and the ratio of the weight of each gene to its bootstrap standard error was used to calculate the Z scores and rank the genes according to their contributions to PLS1[27]. The list of genes showing consistent whole-brain expression pattern with a false discovery rate (FDR) of $p < 0.05$ were fed into an enrichment analysis.

Developmental enrichment evaluated by the significance of overlap between the significant gene list with RNAseq data obtained from BrainSpan dataset (http://www.brainspan. org). The significance was calculated based on Fisher's exact test with FDR correction. The CSEA tool provides simplified results of gene enrichment profiles along six major brain regions (i.e., cortex, thalamus, striatum, cerebellum, hippocampus, amygdala) across 10 developmental periods (from early fetal to young adulthood) approximated from mouse data, yielding a total of 60 combinations of developmental enrichment profiles[25,32].

Gene ontology (GO) enrichment analysis was used to identify enrichment terms through ToppGene (https://toppgene.cchmc.org/)[61]. This includes biological processes (GO: Biological Process), cellular components (GO: Cell Component) and diseases. The significant gene list was included in the enrichment analysis, and the threshold of significance for the enriched term obtained was $p < 0.05$, FDR corrected.

**Statistics and reproducibility**. To control for multiple comparisons, surface-based (20484 vertices) findings were corrected for family-wise errors due to multiple comparisons using a random field theory of $p_{FWE} < 0.05$. Community-wise (seven communities) findings were corrected by FDR correction. In the vertex-wise spatial correlation analysis, spin permutation testing based on spherical rotations (5000 times) was used to assess statistical significant while accounting for spatial autocorrelation[60].In addition, we conducted controlled analyses of important covariates and reproducibility analyses with different sample sizes. We compared gradient eccentricity between male and female TDC, and computed sex × age interaction effect in TDC to assess whether age-related changes are affected by sex effect. We split the Rolandic epilepsy cohort based on AED-medication state (with/without AED-medication: 63/77 cases) and compared gradient eccentricity between AED-medication state, and computed AED × age interaction effect to assess whether the age-related changes are affected by AED. For reproducibility analysis, we carried out a bootstrap analysis in disease × age interaction tests of gradient eccentricity to determine results stability across different sample sizes.

**Reporting summary**. Further information on research design is available in the Nature Portfolio Reporting Summary linked to this article.

## Data availability
Processed data to reproduce our main findings is available on the Open Science Framework (https://osf.io/48tfa/). The processed cortical gene expression was provided by Arnatkevic et al.(https://figshare.com/articles/dataset/AHBAdata/6852911)[57]. The source data behind each figure can be found in Supplementary Data 1.

## Code availability
For functional connectivity gradients, BrainSpace[16] is freely available online: https://github.com/MICA-MNI/brainspace. The codes to calculate gradient eccentricity is available on the Open Science Framework (https://osf.io/48tfa/). The statistics software of Surfstat is freely available online: https://math.mcgill.ca/keith/surfstat.

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

## Acknowledgements

This work was supported by grants of the National Key R&D Program of China (Grant Nos. 2018YFA0701703 and 2020AAA0109505), the National Science and Technology Innovation 2030−Major program of "Brain Science and Brain-Like Research" (2022ZD0211800), Xuzhou Medical University Open Fund Project (XYKF202101), National Natural Scientific Foundation of China (Grant Nos. 82127806, 81871345, 81790653, 81701680).

## Author contributions

Q.Z., J.L., G.L. and Z.Z. contributed to the design of the work; Q.Z., Y.H., and F.Y. contributed to the acquisition of data, Q.Z., J.L. and Q.X. contributed to the analysis of data; Q.Z., J.L., B.B., W.L., G.L. and Z.Z. contributed to the interpretation of data; Q.Z. drafted the work; S.L., B.B., W.L., G.L. and Z.Z. substantively revised the work.

## Competing interests

The authors declare no competing interests.
