## [Peer Review File · Communications Biology]

Reviewers' comments:

Reviewer #1 (Remarks to the Author):

I read with interest the manuscript titled "Atypical development of functional connectome hierarchy in Rolandic epilepsy". The sample size of individuals with rolandic epilepsy and healthy controls is commendable. The methods and statistics are solid and impressive. It reads well and provides some insights into the developmental brain of people with rolandic epilepsy. I only have some concerns from a clinical perspective for authors to consider:

1. I would suggest the authors avoid using "learning disability" in the manuscript to describe the cognitive abnormalities in this type of epilepsy.
2. Children with rolandic epilepsy may display either very frequent centrotemporal spikes or less frequent or even no interictal epileptic activity at all. As the EEG information is not provided, have the authors taken into account the influence of interictal activity? There are enough cases here to do a sub-comparison to see whether there is a difference.
3. The details of AEDs should be provided. AEDs with cognitive side effects (topiramate and zonisamide) may have effects. Mono vs polytherapy could make a difference. I presume that clinicians involved would have avoided using the AEDs with negative cognitive side effects. There are enough cases here to do a sub-comparison to see whether there is a difference between mono vs poly therapy cases as individuals with polytherapy may suggest less well-controlled seizures during the process of the treatment.

Reviewer #2 (Remarks to the Author):

Review comments on atypical development of functional connectome hierarchy in Rolandic epilepsy.

In this resting-state fMRI study, the authors examined the functional connectome hierarchy in epileptic patients by analyzing connectome manifolds. They also examined the gene expression to identify genetic bases to explain such epilepsy-specific atypical functional hierarchy.

Overall, the authors appeared to perform all the analyses in a nice manner. In the meantime, it may be difficult to find biological significance/meanings there. Therefore, the manuscript could become even easier to understand if the authors kindly add some more intuitive explanation on the manifolds eccentricity and put some biological interpretation.

Also, it'd be greatly helpful if the authors address the following methodological concerns.

Major concerns

1. To address the problems of multiple comparisons, the authors used two different methods: FEW-corrected and FDR-corrected in MRI analyses. In principle, such a mixture in one study should be avoided... Given that this study mainly adopted the FWE correction in almost all the statistical tests, it'd be better if the authors show what would happen to the FDR-based results when the statistics are re-evaluated using FEW-based correction.
2. In my understanding, one of the main findings is displayed in Figure 3. However, unfortunately, it is difficult for me to find essential information in it. In particular, the scatter plots in panels D and E seem to need more details. For example, what do the plots in the graphs represent? Participant or the number of ROIs? If it is a participant, which ROI's results are displayed in the plots?
3. Also in Figure 3c and its relevant texts, the author stated the results of "Medication analyses". In my impression, however, it's not medication analyses but mediation analyses... If it is actually medication analysis, the details of the analysis may have to be provided in the text. Otherwise, the

authors may have to give details of the mediation analysis and demonstrate that all the requisite conditions for the analysis are met.

4. The current findings are based on cross-sectional data not on longitudinal ones. Given this, the terms, such as development and deceleration, throughout this manuscript may sound overstating.

5. In the analyses on the gene expression, I am afraid that I cannot find statistically significant links between the gene-based results and MRI-based observations. In my impression, it is necessary for the original claim that the authors provide such quantitative associations between genetic expression and MRI-based manifolds eccentricity.

Reviewer #3 (Remarks to the Author):

Authors study the atypical development of functional connectivity hierarchy in Rolandic epilepsy. They examine the changes of a FC derived metric with development, and its relation to cognition, and genetic factors in 162 patients. This is done by measuring what is referred to as "fMRI multi-axis

functional connectivity manifolds, also described as gradients". They found that "RE is characterized by contraction and deceleration of the functional connectivity manifolds, highlighting the atypical development of connectome hierarchy". These results were also relevant to seizure incidence, cognition and genetic factors.

The paper is solidly written, though there are some methodological issues not clearly explained and the terms that are used are somewhat questionable.

First of all, the term connectome has a clear meaning referring to the structural connectivity, and functional connectome is misleading (moreover, sometimes it is written only connectome, while referring in fact to the FC). Instead the authors should use the term functional connectivity or FC, as it is established.

Second is the usage of the term manifolds. I don't see a justification for calling each projection to the low-dimensional manifold, a manifold itself. The manifold is the low-dimensional object where the activity is contained, but I would restrain for calling its dimensions also manifolds, even in the cases where non-linear decomposition is used, as it is here.

As for the methodological issues:

- l122 "normalized angle matrices". what are those?
- l124 "subject-level connectomes" very misleading as it is FC in fact.
- is there any advantage of using vertices instead of some parcelation?
- Fig. 1: it is not clear how the low-dimensionality reduction is performed. It is said that you start from the FC matrices, but then vertices (not vertexes as stated in l136) are plotted... How is the the diffusion map embedding implemented in fact? This should be explained in intuitive way for any reader, in addition to more rigorous explanation. This is a big conceptual issue.
- ;150 what is functional connectivity density? It seems to be a functional data feature, but it is discussed together with the cortical thickness which is a structural feature.
- what is GO?
- l374 (3) Not very clear how the links here is performed. Is the gene expression information region specific?
- l420 "were overlapped -with- those"
- enrichment analysis needs more elaboration
- literature:
 - there is not much work on the manifold on the whole-brain level, but few recent papers are worth mentioning: Iyer et al Nat Comm 2022 is relevant as a methodologically similar paper that uses dimensionality reduction on whole level brain (although on the nodes, not on the vertices),,, In this papers similarly as here, the nodes are spatial objects, not time points as it is more common in the literature of the dimensionality reduction. It is also worth mentioning the latter, with some recent papers coming out such as: Brown et al Neuroimage 2022, Bolt et al Nat Neurosci 2022, Rué-Queralt et al Comm Biol 2021.
 - the current paper does not identify casualty between the observed phenomena on the manifold and the developmental and genetic factors, since this would require a mechanistic model. However, it is worth at least discussing that that kind of approaches exist; for example, Courtiol et

al J Neurosci 2020 for a resting state of temporal lobe epilepsy, but also Jirsa's group has many other papers on modelling seizure propagation.
More general TVB as a platform integrated in EBRAINS allows constructing such mechanistic models, e.g. see Schirner et al Neuroimage 2022.

Reviewer 1:

1. I would suggest the authors avoid using “learning disability” in the manuscript to describe the cognitive abnormalities in this type of epilepsy.

RE: Thank you for the suggestion. We agree that the term “learning disability” may not be the most appropriate description for the cognitive abnormalities observed in Rolandic epilepsy. We have revised the manuscript to avoid using this term and have instead used more specific descriptions of the cognitive deficits observed in our study (L31). Thank you for bringing this to our attention.

2. Children with rolandic epilepsy may display either very frequent centrotemporal spikes or less frequent or even no interictal epileptic activity at all. As the EEG information is not provided, have the authors taken into account the influence of interictal activity? There are enough cases here to do a sub-comparison to see whether there is a difference.

RE: Thank you for your insightful question. We acknowledge that interictal epileptic activity, which may vary in frequency and duration across individuals with Rolandic epilepsy, may affect functional connectivity and cognitive abilities. In our study, we collected detailed clinical information on each individual’s seizure incidence. We did perform sub-analyses on our sample to examine the relationship between the seizure free duration and manifold alterations. We found that individuals with more seizure free duration had better functional connectivity manifolds/gradients, suggesting that seizure activity may be an important factor contributing to the atypical connectivity hierarchy observed in Rolandic epilepsy.

We agree that future studies incorporating EEG data would be valuable for better understanding the relationship between interictal activity and connectivity gradients in Rolandic epilepsy. A subset of children in our database performed simultaneous EEG-fMRI (n = 48), and it can be used to monitor for significant intermittent epileptic activity during the scan (30 patients with discharge and 18 without discharge). We added an experiment to the supporting material to compare whether there is a difference between patients with and without discharge during the scan. The result showed there is no significant difference between two sub-groups (L195). Detailed methods and results have been added to the supplementary material (Supplementary Methods 2 and Supplementary Result 8).

This may be due to the fact that epileptic discharges in RE patients are usually short and highly variable in number, which is not sufficient to cause gradients changes throughout the scan period. Interestingly we are conducting another study focusing on dynamic gradients changes during discharges in patients with absence epilepsy.

3. The details of AEDs should be provided. AEDs with cognitive side effects (topiramate and zonisamide) may have effects. Mono vs polytherapy could make a

difference. I presume that clinicians involved would have avoided using the AEDs with negative cognitive side effects. There are enough cases here to do a sub-comparison to see whether there is a difference between mono vs poly therapy cases as individuals with polytherapy may suggest less well-controlled seizures during the process of the treatment.

RE: Thank you for your question and comments. We agree that the some of the antiepileptic drugs (AEDs) could have an impact on cognitive abilities and functional connectivity. In our study, we collected information on the AEDs used by each patients, and also test AED effect on age-related eccentricity change in RE. The results showed significant lower eccentricity in DMN and FPN of AED-medicated patients. But no significant AED \times age interaction effect was observed when estimating the possible AED effect on age-related eccentricity change trends. Since the main purpose of this paper was to explore the age-related change of the patient's eccentricity, we included the use of AEDs as a covariate in the main analysis. However, no further analysis of the effect of specific drug use on the eccentricity was carried out.

It is worth noting that many of the commonly prescribed AEDs have limited cognitive side effects or may even have positive cognitive effects. Most of our patients were treated with valproic acid, levetiracetam and oxcarbazepine, with only 2 patients on medications that may affect cognition. The details were as follows: 41 patients were treated with monotherapy (levetiracetam: 21, valproic acid: 14, oxcarbazepine: 7, Lamotrigine: 1), 22 patients were treated with polytherapy (levetiracetam + oxcarbazepine: 9, levetiracetam + valproic acid: 4, levetiracetam + Lamotrigine: 5, valproic acid + Carbamazepine: 1, valproic acid + Phenobarbital: 1, Lamotrigine + Topiramate: 1, oxcarbazepine + Lamotrigine: 1). This information was added to the supplementary material (Supplementary Result 7).

Reviewer 2:

1. In this resting-state fMRI study, the authors examined the functional connectome hierarchy in epileptic patients by analyzing connectome gradients. They also examined the gene expression to identify genetic bases to explain such epilepsy-specific atypical functional hierarchy.

Overall, the authors appeared to perform all the analyses in a nice manner. In the meantime, it may be difficult to find biological significance/meanings there. Therefore, the manuscript could become even easier to understand if the authors kindly add some more intuitive explanation on the gradients eccentricity and put some biological interpretation.

RE:

Thank you for your feedback and comments. An important aspect of the development of brain networks in childhood is the separation and integration of brain functions. In previous studies, a trend of expansion with age was found in principal gradients of the functional and structural network. It is thought

to represent the development of segregation of brain functions. In the present study we found a tendency for all three major gradients to expand with age and designed the metric of eccentricity to synthetically quantify the degree of multidimensional gradient separation in brain regions. Then we confirmed its linear relationship between eccentricity and age, it means that this metric can be integrated to show the topography of the separation trend of the three main gradients with age development.

Using the good properties of this metric, we identified atypical development of the connectome hierarchy in patients with Rolandic epilepsy, characterized by contraction and deceleration of the functional connectome manifold, which is associated with seizure incidence, cognition, and connectivity deficit, and development-associated genetic basis. We discuss the biological significance of the index further in the introduction(L66) and discussion(L271), which we hope will help the reader understand.

2. To address the problems of multiple comparisons, the authors used two different methods: FEW-corrected and FDR-corrected in MRI analyses. In principle, such a mixture in one study should be avoided... Given that this study mainly adopted the FWE correction in almost all the statistical tests, it'd be better if the authors show what would happen to the FDR-based results when the statistics are re-evaluated using FEW-based correction.

RE: Thank you for your comment and suggestion.

In our study, we chose the appropriate correction for different statistics. Specifically, we used FWE-corrected methods for the surface-based analysis (number of comparisons: 20484). This correction uses random-field theory to perform multiple comparison correction on the potential clusters generated after the initial threshold. The biggest advantage of random-field theory correction is the introduction of smoothness for judgment. This is the default correction in SPM (volume-based) and Surfstat (surface-based analysis). And in the network level analysis (number of comparisons: 7), we directly used the conventional FDR correction. We have added a section (Statistics and reproducibility, L483) to the methods that presents the different correction methods and the rationale for their choice in a unified manner.

3. In my understanding, one of the main findings is displayed in Figure 3. However, unfortunately, it is difficult for me to find essential information in it. In particular, the scatter plots in panels D and E seem to need more details. For example, what do the plots in the graphs represent? Participant or the number of ROIs? If it is a participant, which ROI's results are displayed in the plots?

RE: Thank you for your comment and question. We apologize for the lack of clarity in Figure 3. In Figure 3, Scatter plots of D and E depict the relationship between manifold eccentricity and seizure or cognition, where the eccentricity values are derived from the average of intra DMN. Each point in the graph represents a subject. We have modified Figure 3 and its legend to be

more easily understood.

4. Also in Figure 3c and its relevant texts, the author stated the results of “Medication analyses”. In my impression, however, it’s not medication analyses but mediation analyses… If it is actually medication analysis, the details of the analysis may have to be provided in the text. Otherwise, the authors may have to give details of the mediation analysis and demonstrate that all the requisite conditions for the analysis are met.

RE: Thank you for bringing this to our attention. You are correct that there was a typo in Figure 3c and it should be “Mediation Analyses” instead of “Medication Analyses”. We apologize for any confusion this may have caused. We will make sure to correct the typo in the manuscript and provide a clear description of the mediation analysis in the text.

5. The current findings are based on cross-sectional data not on longitudinal ones. Given this, the terms, such as development and deceleration, throughout this manuscript may sound overstating.

RE: We appreciate the reviewer’s comment and agree that the terms “development” and “deceleration” may sound overstating in a cross-sectional study. We apologize for any confusion this may have caused. To address this concern, we will revise the language throughout the manuscript to clarify that the observed differences in functional connectome hierarchy are indicative of age-related alteration in Rolandic epilepsy, rather than definitive evidence of developmental changes. We will also highlight the limitations of cross-sectional designs and the need for future longitudinal studies to confirm these findings and investigate the potential changes in connectome hierarchy over time (L342).

6. In the analyses on the gene expression, I am afraid that I cannot find statistically significant links between the gene-based results and MRI-based observations. In my impression, it is necessary for the original claim that the authors provide such quantitative associations between genetic expression and MRI-based gradients eccentricity.

RE: Thank you for your question. genetic factors play important roles in brain connectomes. Of course, it would be most beneficial if the relationship between brain expression and macroscopic imaging could be obtained directly at the patient level. However, in most cases, we do not have access to brain tissue gene expression data from patients with benign epilepsy.

In such cases, The Allen Human Brain Atlas (AHBA) microarray dataset has been used to identify transcriptomes associated with human neuroimaging with multimodal evidence suggesting a link between conserved gene expression and functionally relevant circuitry. This analysis was accomplished by spatial correlation of altered patterns of MRI-based manifolds eccentricity with brain expression profiles. Combining neuroimaging and gene transcripts has provided

insight transcripts has provided insight into how disease-related alterations at the microscale architecture drive macroscale brain abnormalities in manifold development¹, various mental disorders²³, and epilepsy⁴ in recent years.

We used the Allen Human Brain Atlas to obtain gene expression information in specific regions of the brain, and then linked this information to age-related alterations of gradient eccentricity in children with Rolandic epilepsy from the same regions. This strategy is used to see which genes have spatial expression patterns that are significantly correlated with the image phenotype. Specifically, Partial least squares regression (PLS) was used to determine the relationship between hierarchical age-related alterations (mean t-values of disease \times age interaction of gradient eccentricity from 150 cortical regions in the left hemisphere) and transcriptional activity for all 10,027 genes in 150 cortical regions. We have further modified the results and methods section to make it easier for the reader to understand.

As the reviewer stated that this is a cross-sectional study, we further used this spatial transcriptomic association analysis and various types of enrichment analysis to further explain the relationship between age-related eccentricity changes and development. The results showed significant associations with multiple structural development-related genes, especially in late childhood-to-adolescence time window, which consist with Rolandic epilepsy related brain region and self-limited natural history. We did under-represent the motivation and discussion of this part of the analysis. We have further clarified it in the manuscript and also the limitations of this linkage. References:

1. Park BY, Bethlehem RA, Paquola C, *et al.* An expanding manifold in transmodal regions characterizes adolescent reconfiguration of structural connectome organization. *eLife*. Mar 31 2021
2. Li J, Seidlitz J, Suckling J, *et al.* Cortical structural differences in major depressive disorder correlate with cell type-specific transcriptional signatures. *Nature communications*. Mar 12 2021;12(1):1647
3. Hettwer, M.D., Larivière, S., Park, B.Y. et al. Coordinated cortical thickness alterations across six neurodevelopmental and psychiatric disorders. 2022; *Nature communications*. 13, 6851
4. Larivière S, Royer J, Rodríguez-Cruces R. et al. Structural network alterations in focal and generalized epilepsy assessed in a worldwide ENIGMA study follow axes of epilepsy risk gene expression. *Nature communications*. 2022 Jul 27;13(1):4320.

Reviewer #3 (Remarks to the Author):

1. Authors study the atypical development of functional connectivity hierarchy in Rolandic epilepsy. They examine the changes of a FC derived metric with development, and its relation to cognition, and genetic factors in 162 patients.

This is done by measuring what is referred to as “fMRI multi-axis functional connectivity manifolds, also described as gradients”. They found that “RE is characterized by contraction and deceleration of the functional connectivity manifolds, highlighting the atypical development of connectome hierarchy”. These results were also relevant to seizure incidence, cognition and genetic factors.

The paper is solidly written, though there are some methodological issues not clearly explained and the terms that are used are somewhat questionable.

First of all, the term connectome has a clear meaning referring to the structural connectivity, and functional connectome is misleading (moreover, sometimes it is written only connectome, while referring in fact to the FC). Instead the authors should use the term functional connectivity or FC, as it is established.

RE: Thank you for your comment on the paper. When we talk about connectome, most of the time it is about structural connectivity, but I also believe that functional connectivity is also a way to describe connectome. To avoid confusion, I am happy to use the term functional connectivity and have modified it throughout the text.

2. Second is the usage of the term manifolds. I don't see a justification for calling each projection to the low-dimensional manifold, a manifold itself. The manifold is the low-dimensional object where the activity is contained, but I would restrain for calling its dimensions also manifolds, even in the cases where non-linear decomposition is used, as it is here.

RE: Thank you for your comment. I agree with you that the usage of the term “manifolds” may be confusing in this context. The term gradient is more commonly used in the decomposition of brain functional connectivity matrices, and I will use the term gradient throughout the text instead.

3. As for the methodological issues:

- 1122 “normalized angle matrices”. what are those?

RE: In the context of the manuscript, “normalized angle matrices” refer to cosine distance between all pairs of rows to estimate the similarity in connectivity patterns between each pair of voxels, thereby obtaining a symmetrical similarity matrix. We describe it in more detail in the method(L402).

4. - 1124 “subject-level connectomes” very misleading as it is FC in fact.

RE: Thanks for your comments, we have adopted the term functional connectivity throughout the text.

5. - is there any advantage of using vertices instead of some parcellation?

RE: Adult brain parcellation are more difficult to adapt to the rapidly developing brains of children. Recent study focusses on development of

connectivity gradient also used the vertices analysis (references 22, 24 in main body). Moreover, the main disease we are dealing with is epilepsy, and the altered brain function caused by epileptic activity may be different from using a brain template created in a normal person. If a template is used it may mask some of the alterations in areas associated with epileptic activity. Because vertex-based analysis can be more sensitive to changes in local connectivity patterns, we base our results primarily on the vertices, but we also use partitioning of the resting-state brain network to further refine the results and improve interpretability.

6. - Fig. 1: it is not clear how the low-dimensionality reduction is performed. It is said that you start from the FC matrices, but then vertices (not vertexes as stated in 1136) are plotted... How is the diffusion map embedding implemented in fact? This should be explained in intuitive way for any reader, in addition to more rigorous explanation. This is a big conceptual issue.

RE: We applied diffusion map embedding, a nonlinear manifold learning approach, to identify gradient components explaining connectivity variance in descending order (each of $1 \times 20,484$). In brief, the algorithm estimates a low-dimensional embedding from a high-dimensional connectivity matrix. In this space, cortical vertices that are strongly interconnected by either many connections or few very strong connections are closer together, whereas vertices with only little or no inter-connectivity are farther apart. The name of this approach, which belongs to the family of graph Laplacians, derives from the equivalence of the Euclidean distance between points in the embedded space and the diffusion distance between probability distributions centered at those points. Compared to other non-linear manifold learning techniques, the diffusion maps algorithm is relatively robust to noise and computationally inexpensive. Notably, the algorithm is controlled by a single parameter α , which controls the influence of density of sampling points on the manifold ($\alpha = 0$, maximal influence; $\alpha = 1$, no influence). In this study, we followed previous recommendation and set $\alpha = 0.5$, a choice that retains the global relations between data points in the embedded space.

Diffusion embedding has been widely used in functional connectivity gradients and has been well described by many literatures. We will describe the specific methods in more detail (L402).

7. - ;150 what is functional connectivity density? It seems to be a functional data feature, but it is discussed together with the cortical thickness which is a structural feature.

RE: Functional connectivity density (FCD) refers to the amount of functional connectivity between brain regions. It is a measure of the strength of connections between brain regions in the FC matrices. We calculated global FCD mapping was calculated to measure the vertex-wise functional connectivity strength ¹. The number of functional connections was determined through Pearson

correlations between time-varying signals of a vertex and those in other vertex using an arbitrary threshold $r = 0.3$ (detailed in Supplementary Materials and methods 1).

FCD and cortical thickness, have been used in for study functional and structural brain development in children in many previous studies. To determine the unique contribution of eccentricity versus FCD and cortical thickness to age, we ran robust linear regression on eccentricity controlling for FCD or cortical thickness. we found that eccentricity had independent age predictive efficiency.

8. - what is GO?

RE: GO stands for Gene Ontology, which is a standardized system for annotating genes and their functions in different organisms. It provides a set of structured, controlled vocabularies for describing gene products in terms of their associated biological processes, molecular functions, and cellular components. The GO project is an international effort to unify the representation of gene and gene product attributes across all species, and to facilitate functional genomics research. We have given the full name in the text.

9. - 1374 (3) Not very clear how the links here is performed. Is the gene expression information region specific?

RE: Yes, the gene expression information is region-specific. We used the Allen Human Brain Atlas to obtain gene expression information in specific regions of the brain, and then linked this information to age-related alterations of gradient eccentricity in children with Rolandic epilepsy from the same regions. This strategy is used to see which genes have spatial expression patterns that are significantly correlated with the image phenotype. Specifically, Partial least squares regression (PLS) was used to determine the relationship between hierarchical age-related alterations (mean t-values of disease \times age interaction of gradient eccentricity from 150 cortical regions in the left hemisphere) and transcriptional activity for all 10,027 genes in 150 cortical regions. We have further modified the results and methods section to make it easier for the reader to understand.

10. - 1420 “were overlapped -with- those”

RE: Thank you for the heads up, we have double checked the grammar of the entire text.

11. - enrichment analysis needs more elaboration

RE: Thank you for your suggestion, both enrichment analyses are based on gene list enrichment analysis and are implemented through an interactive web tool. We add further details of the analyses in the Methods(L470).

12.- literature:- there is not much work on the manifold on the whole-brain level, but few recent papers are worth mentioning: Iyer et al Nat Comm 2022 is relevant as a methodologically similar paper that uses dimensionality reduction on whole level brain (although on the nodes, not on the vertices),,, In this papers similarly as here, the nodes are spatial objects, not time points as it is more common in the literature of the dimensionality reduction. It is also worth mentioning the latter, with some recent papers coming out such as: Brown et al Neuroimage 2022, Bolt et al Nat Neurosci 2022, Rué-Queralt et al Comm Biol 2021.

RE: Thank you very much for your suggestion and we have again reviewed the relevant research developments in recent times and added them to the text. It is a great pity that we have not been able to locate some of the literature you have provided. The issue of whole-brain vertex level analysis was addressed in detail in our previous response.

13. - the current paper does not identify casualty between the observed phenomena on the manifold and the developmental and genetic factors, since this would require a mechanistic model. However, it is worth at least discussing that that kind of approaches exist; for example, Courtiol et al J Neurosci 2020 for a resting state of temporal lobe epilepsy, but also Jirsa' s group has many other papers on modelling seizure propagation.

More general TVB as a platform integrated in EBRAINS allows constructing such mechanistic models, e.g. see Schirner et al Neuroimage 2022.

RE:

It is very exciting to use some mechanistic model to elucidate some causal relationships, but how to relate gradient data to the virtual brain is a question that needs further research. Thank you very much for the direction you provided and we will try to incorporate this experimental paradigm in future research. In the meantime we have added a discussion about this to the discussion(L349).

REVIEWERS' COMMENTS:

Reviewer #1 (Remarks to the Author):

The authors answered all my concerns.

Reviewer #2 (Remarks to the Author):

The authors have appeared to respond to all my concerns.

Reviewer #3 (Remarks to the Author):

Authors have addressed my comments and I endorse publishing the manuscript.